 eLIFE

# Midbrain dopamine neurons compute inferred and cached value prediction errors in a common framework

Brian F Sadacca[1]*, Joshua L Jones[1], Geoffrey Schoenbaum[1,2,3]*

[1]Intramural Research program of the National Institute on Drug Abuse, National Institutes of Health, Bethesda, United States; [2]Department of Anatomy and Neurobiology, University of Maryland School of Medicine, Baltimore, United States; [3]Department of Neuroscience, Johns Hopkins School of Medicine, Baltimore, United States

**Abstract** Midbrain dopamine neurons have been proposed to signal reward prediction errors as defined in temporal difference (TD) learning algorithms. While these models have been extremely powerful in interpreting dopamine activity, they typically do not use value derived through inference in computing errors. This is important because much real world behavior – and thus many opportunities for error-driven learning – is based on such predictions. Here, we show that error-signaling rat dopamine neurons respond to the inferred, model-based value of cues that have not been paired with reward and do so in the same framework as they track the putative cached value of cues previously paired with reward. This suggests that dopamine neurons access a wider variety of information than contemplated by standard TD models and that, while their firing conforms to predictions of TD models in some cases, they may not be restricted to signaling errors from TD predictions.

*For correspondence: brian. sadacca@nih.gov (BFS); geoffrey. schoenbaum@nih.gov (GS)

**Competing interests:** The authors declare that no competing interests exist.

## Introduction

Midbrain dopamine neurons have been proposed to signal the reward prediction errors defined in temporal difference (TD) learning algorithms (*Schultz et al., 1997*; *Sutton, 1988*). This proposal was initially based on observations that these neurons fired more strongly to unpredicted than to predicted reward, suppressed firing on omission of a predicted reward, and developed firing to reward-paired cues with learning (*Mirenowicz and Schultz, 1994*). Further work has shown that phasic activity in dopamine neurons obeys formal predictions for such TD error signals under more complex conditions (*Waelti et al., 2001*; *Tobler et al., 2003*; *Lak et al., 2014*; *Pan et al., 2005*; *Hart et al., 2014*; *Bayer and Glimcher, 2005*; *Hollerman and Schultz, 1998*), including in tasks such as blocking and conditioned inhibition, in which experimental conditions are arranged to precisely distinguish between prediction error signals and other possible explanations of such activity. These studies have confirmed that the neural correlates correspond closely to the theoretical accounts. Indeed careful work in monkeys has shown that the activity provides a quantitative match with the error signal described in the TD model (*Bayer and Glimcher, 2005*). With the advent of optogenetic techniques and Cre-driver lines in rats, it has also been shown in rats that artificially stimulating or inhibiting likely dopamine neurons for very brief periods is sufficient to restore the associative learning driven by endogenous positive or negative prediction errors (*Steinberg et al., 2013*; *Chang et al., 2016*), suggesting that phasic activity in dopamine neurons can act like a prediction error, at least in some downstream targets and behavioral paradigms (*Glimcher, 2011*; *Schultz, 2002*).

**eLife digest** Learning is driven by discrepancies between what we think is going to happen and what actually happens. These discrepancies, or 'prediction errors', trigger changes in the brain that support learning. These errors are signaled by neurons in the midbrain – called dopamine neurons – that fire rapidly in response to unexpectedly good events, and thereby instruct other parts of the brain to learn about the factors that occurred before the event. These events can be rewards, such as food, or cues that have predicted rewards in the past.

Yet we often anticipate, or infer, rewards even if we have not experienced them directly in a given situation. This inference reflects our ability to mentally simulate likely outcomes or consequences of our actions in new situations based upon, but going beyond, our previous experiences. These inferred predictions of reward can alter error-based learning just like predictions based upon direct experience; but do inferred reward predictions also alter the error signals from dopamine neurons?

Sadacca et al. tested this question by exposing rats to cues while recording the activity of dopamine neurons from the rats' midbrains. In some cases, the cues directly predicted rewards based on the rats' previous experience; in other cases, the cues predicted rewards only indirectly and based on inference. Sadacca et al. found that the dopamine neurons fired in similar ways in response to the cues in both of these situations. This result is consistent with the proposal that dopamine neurons use both types of information to calculate errors in predictions. These findings provide a mechanism by which dopamine neurons could support a much broader and more complex range of learning than previously thought.

But is it a TD prediction error? The errors in TD models, at least as they have been applied to interpret the firing of dopamine neurons, are based on the value that has been assigned to events based on direct experience (*Glimcher, 2011*; *Schultz, 2002*; *Niv and Schoenbaum, 2008*; *Clark et al., 2012*). This so-called *cached* value is pre-computed and resides in the predictive event, be it a primary reward or reward-predicting cue. The value is calculated free of any other predictive information about the environment, defining this class of algorithms as *model-free*. While these TD models have been extremely powerful in interpreting the activity of dopamine neurons in tasks in which value is based on experienced reward, they are unable to compute prediction errors elicited in situations in which value is derived through inference rather than through direct experience. Deriving a value (or a prediction) based on inference – meaning to deduce from an understanding of the relationships amongst environmental stimuli and events - is a hallmark of a second class of algorithms, termed *model-based* (*Daw et al., 2005*).

This distinction between algorithms is important because much of our real world behavior, and thus many of our opportunities for error-driven learning, is based on this model-based inference (*Daw et al., 2005*; *Doll et al., 2012*). Rarely does one engage in value-based behavior that is simply a pure repetition of prior learning; typically moderating or mitigating information acquired in other situations, separate from the original learning, influences our decision-making. This basic concept is operationalized (and the effect of such *inferred* or *model-based* value, isolated) in sensory pre-conditioning (*Brogden, 1939*). In this task, animals first learn that one innocuous cue (cue A) predicts another (cue B), in the absence of reward, and then later learn that the second cue (B) is a reliable predictor of reward. Cue A has not been directly paired with reward and thus it has not had any opportunity to acquire any cached value (*Glimcher, 2011*; *Niv and Schoenbaum, 2008*). As a result, cue A cannot elicit a TD prediction error, despite the fact that it has value as defined by the animal's responding and its ability to modulate error-driven learning (*Jones et al., 2012*). If TD models are an accurate and complete description of the information contained in dopamine neuron activity, then cue A should not elicit dopamine neuron firing, at least not above the level of a control cue.

## Results

To test this, we trained 14 rats with recording electrodes implanted in the ventral tegmental area (VTA) in a sensory-preconditioning task. In the first phase, rats learned to associate two pairs of

environmental cues (A->B; C->D) in the absence of reward. As there was no reward, rats showed no significant responding at the food cup and no differences in responding during the different cues (ANOVA, $F_{3, 55}$ = 0.7, p=0.52; *Figure 1A*). In the second phase, rats learned that the second cue of one pair (B) predicted reward and the other (D) did not; learning was reflected in an increase in responding at the food cup during presentation of B (ANOVA, main effect of cue: $F_{1, 163}$ = 280.1, p<0.001, main effect of session: $F_{5, 163}$ = 9.7, interaction: $F_{5, 163}$ = 10.81, p<0.001; *Figure 1B*). Finally, in the third phase, the rats were presented again with the four cues, first a reminder of cue B and D's reward contingency followed by an unrewarded probe test of responding to cues A and C. As expected, the rats responded at the food cup significantly more during presentation of A, the cue that predicted B, than during presentation of C, the cue that predicted D (ANOVA, main effect of cue: $F_{1, 167}$ = 8.7, p<0.001, main effect of trial: $F_{5, 167}$ = 6.08, p<0.001, interaction: $F_{5, 167}$ = 2.07, p=0.07; *Figure 1C*).

Single unit activity was recorded in the VTA throughout training. To identify putative dopamine neurons, we used a recently developed, optogenetically-validated strategy that classifies VTA neurons on the basis of their response dynamics during Pavlovian conditioning. In published work (*Cohen et al., 2012*; *Eshel et al., 2015*), this strategy identified VTA dopamine neurons (i.e. neurons expressing Cre under the control of the promoter for the dopamine transporter) with near perfect fidelity. Here we applied this same analysis to the mean normalized responses of all VTA neurons recorded during conditioning and reminder sessions (n = 632; *Figure 2A*). We extracted the major modes of variation among the neurons with principal components analysis (PCA; *Figure 2B*) and then performed hierarchical clustering on those PCs (*Figure 2C*). This analysis successfully extracted the 3 previously described VTA response types from our data (*Figure 2D*): neurons with sustained excitation to cues and reward (putative GABAergic), neurons with phasic excitation to cue onset and reward onset (putative dopaminergic), and neurons with sustained inhibition to cue and reward (unknown). We then assessed the responses of the putative dopamine neurons (n = 304) to the cue and reward over the course of conditioning. Consistent with their classification, we found changes in firing during conditioning that were in accord with signaling of reward prediction errors. Specifically, early in conditioning, these neurons' maximal response occurred just after reward delivery

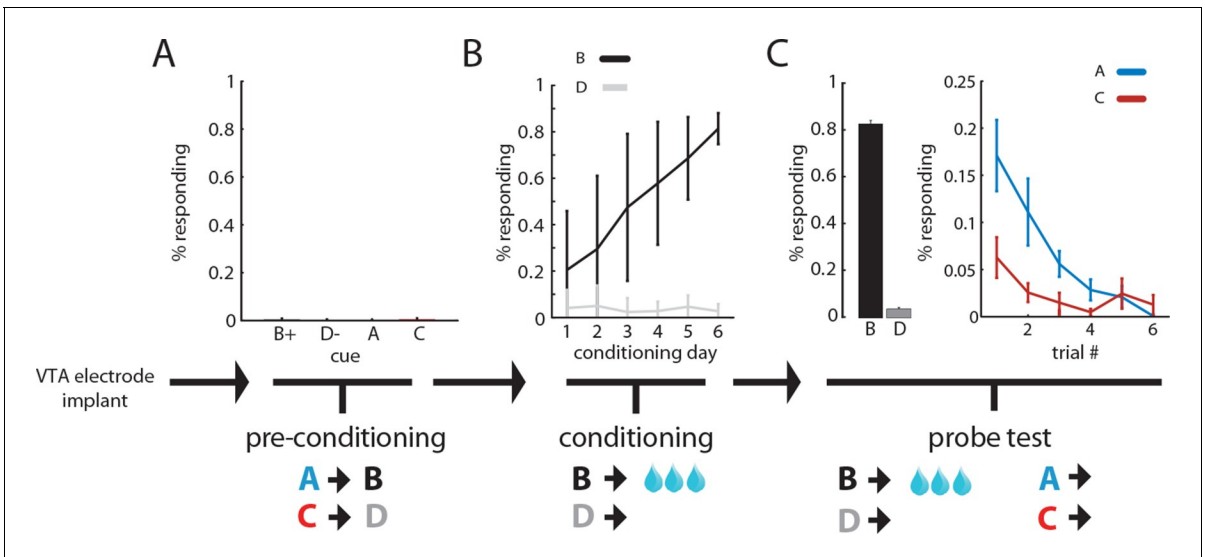

**Figure 1.** Rats infer the value of cues during sensory preconditioning. Panels illustrate the task design and show the percentage of time spent in the food cup during presentation of the cues during each of the three phases of training. In the 'preconditioning' phase (**A**) rats learn to associate auditory cues in the absence of reinforcement; during this phase there is minimal food cup responding (ANOVA, F (3, 55) = 0.7, p = 0.52). In subsequent 'conditioning' (**B**), rats learn to associate one of the cues (B) with reward; conditioned responding at the food cup during B increases across sessions (ANOVA, main effect of cue: F (1, 163) = 280.1, p<0.001, main effect of session: F (5, 163) = 9.7, interaction: F (5, 163) = 10.81, p<0.001). In a final 'probe' test (**C**), rats are presented with each of the 4 auditory cues; conditioned responding at the food cup is maintained to B and is also now evident during presentation of A, the cue that had been paired with B in the preconditioning phase (ANOVA, main effect of cue: F (1, 167) = 8.7, p<0.001, main effect of trial: F (5, 167) = 6.08, p<0.001, interaction: F (5, 167) = 2.07, p=0.07).

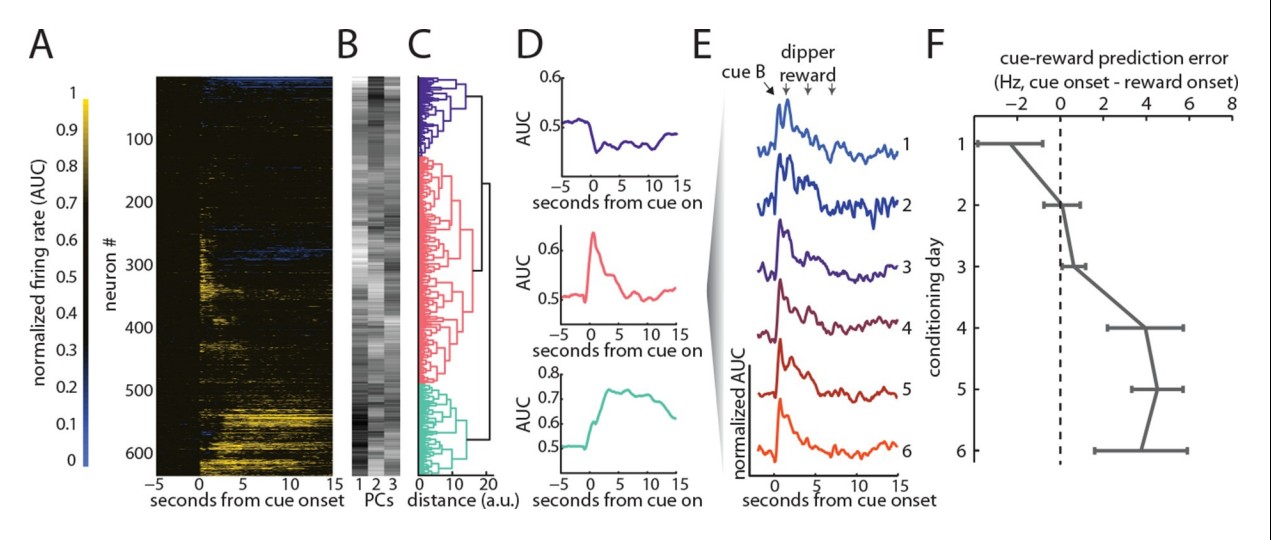

**Figure 2.** VTA dopamine neurons exhibit firing to a reward-paired cue that is consistent with TD error signaling. We recorded 632 neurons across all days of conditioning and the final reminder session. (**A**) Normalized responses (AUC) are displayed for each neuron, sorted by the classification algorithm applied by Cohen, Uchida and colleagues (**Cohen et al., 2012**). The first three principal components (PCs) were extracted, to find the major modes of this population's response (**B**), then hierarchical agglomerative clustering was used on those PCs to identify similar neural responses; groups identified are highlighted in color (**C**); The mean group response of each of the populations identified are displayed (**D**); in accordance with previous results (**Cohen et al., 2012**) we found populations undergoing sustained excitation, phasic excitation, and sustained inhibition. Consistent with identification as putative dopamine neurons, the average (AUC) response to cue B from the phasic group on each day of conditioning exhibited a peak response that was highest to reward early in conditioning and migrated to earlier cue onset across conditioning (**E–F**, r(302) = 0.24, p<0.01). This change in firing is in accordance with signaling of a TD error.

(**Figure 2E**, top trace), whereas late in conditioning, the maximal response occurred just after onset of the cue predicting that reward (**Figure 2E**, lower traces). As a result, the difference in activity at the time of cue onset versus reward increased significantly from the start to the end of conditioning ($r_{302}$ = 0.24, p<0.01; **Figure 2F**), consistent with signaling of TD prediction errors (**Glimcher, 2011**; **Schultz, 2002**).

Having established that putative dopamine neurons identified in this manner exhibit firing during conditioning consistent with signaling of TD errors, we next examined activity in neurons recorded in just the probe test. We again identified these neurons by their pattern of firing to the reward predictive cue (n = 102; **Figure 3A–D**). As before, this analysis identified a group of cells with strong phasic responses to B, the cue that had been directly paired with reward (n = 52). While this response generalized somewhat to D, the control cue that had been presented without reward during conditioning sessions, these neurons fired significantly more during the first second of B than to D ($t_{51}$ = 4.40, p<0.001, black versus gray lines, respectively, with shading for SEM, **Figure 3E**).

However, in addition to this expected pattern of firing, these cells also had strong phasic responses to both preconditioned cues (blue and red traces, A and C, respectively, with shading for SEM, **Figure 3E**). While the common element of these responses could reflect novelty or salience, since these cues had not been presented for a number of days, or perhaps generalization from conditioning to B, the actual phasic neural response was significantly stronger for A, the cue that predicted the reward-paired cue, than for C, the preconditioned control cue ($t_{51}$ = 5.02, p<0.001). This difference cannot be explained on the basis of novelty, salience, or generalization, since A and C were treated similarly. Nor can it be explained by direct experience with reward, because A was never paired with reward, and it was only paired with B before conditioning. Thus, the phasic response in these putative dopamine neurons appeared to be influenced by inferred value of cue A. Interestingly, the neural response was perhaps somewhat better at discriminating A from C (**Figure 3E**, bottom panel) than B from D (**Figure 3E**, top panel), perhaps reflecting the differences in training between A and C, which were only presented a few times in unrewarded sessions, versus B and D, which were presented many times across several days of conditioning. Despite this, the

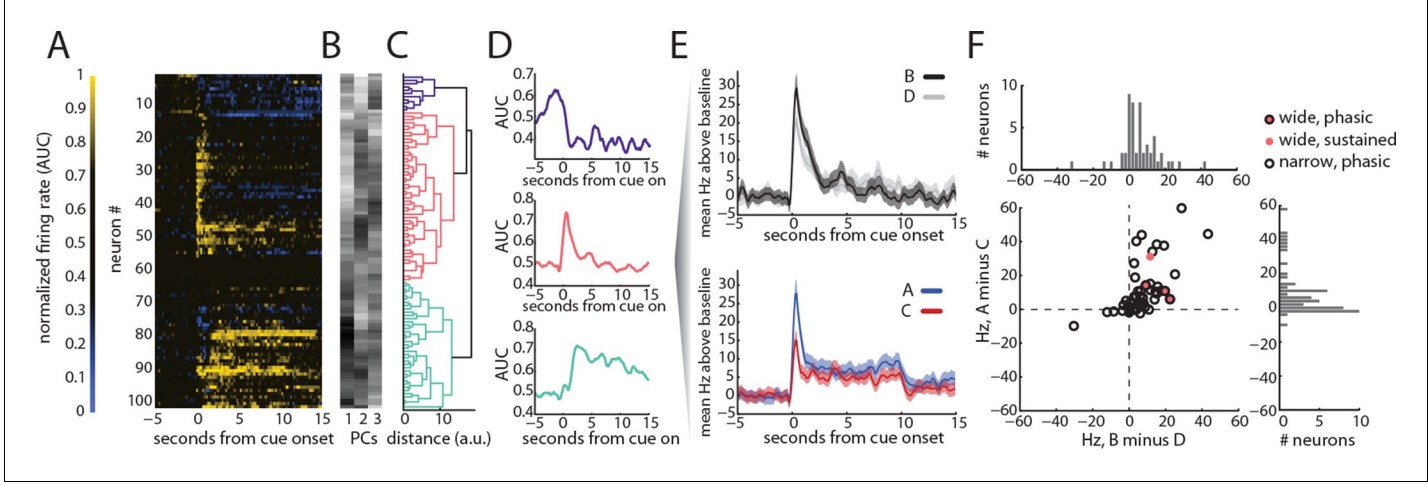

**Figure 3.** VTA dopamine neurons exhibit firing to a pre-conditioned cue that is not consistent with TD error signaling. We recorded 102 neurons during the probe test. AUC normalized neural responses were classified with a hierarchical clustering as in *Figure 2* (A–D) in order to identify putative dopamine neurons (n = 52). In addition, we also identified 4 neurons based on traditional waveform criteria. While the classified putative dopamine neurons showed firing to all cues, they exhibited the largest responses at the onset of B, the reward-paired cue (significantly above responding to D, t (51) = 4.40, p<0.001), and to A, the cue that had been paired with B in the preconditioning phase (significantly above responding to control cue C, t (51) = 5.02, p<0.001) (E–F). Further, the activity elicited by these two cues was strongly correlated (F), suggesting that dopamine neurons code errors elicited by these two types of cues in a common framework (correlation between B–D and A–C, r (50) = 0.63, p<0.001).

The following figure supplements are available for figure 3:

**Figure supplement 1.** Neural responses from phasic and tonic wide-waveform neurons.

**Figure supplement 2.** Neural responses from 39 neurons classified as tonically excited by cue B.

**Figure supplement 3.** Neural responses from 11 neurons classified as tonically inhibited by cue B.

influence of the inferred value of cue A on the firing of these neurons during the first second of their cue response was strongly and significantly correlated with the influence of value on these neurons firing at the onset of B, the cue directly paired with reward ($r_{50}$ = 0.63, p<0.001; *Figure 3F*). Notably this was also true for a handful of neurons (n = 4) that exhibited the classic wide, polyphasic waveforms traditionally used to identify dopamine neurons (*Figure 3F*, filled circles, see *Figure 3—figure supplement 1* for PSTH's). This relationship in the initial phasic response to the cues did not reach significance in the other two neural subtypes identified by the clustering analysis (see *Figure 3—figure supplements 2* and *3* for analyses of tonically-modulated neurons).

Beyond their phasic responses at the start of the cues and to reward, the putative dopamine neurons also exhibited another notable feature: the average response of these neurons throughout cues A and C was above baseline, and this sustained firing was significantly higher to cue A than C (*Figure 3E*, final 9s of cues, $t_{51}$ = 2.56, p<0.05). This elevated firing may be a sign of dopamine's reported ability to anticipate proximity to reward or to signal state value (*Howe et al., 2013*; *Hamid et al., 2016*), if our rats' expectation of reward delivery is based on knowing that progression to the offset of cue A should lead to the subsequent presentation of cue B and then reward. Importantly, in our design, reward is presented during B rather than at its termination. This would explain why this pattern of sustained firing is not present throughout cue B (*Figure 3E*, $t_{51}$= 1.09, p=0.278). Interestingly this pattern of sustained and differential firing to A (vs C) and not to B (vs D) in the putative dopamine neurons is the mirror image of firing in neurons classified as tonically excited, which showed relatively modest changes in sustained firing to A and C and much larger increases in firing to B (see *Figure 3—figure supplement 2*). This relationship would be consistent with recent proposals that these neurons, thought to be GABAergic (*Cohen et al., 2012*), exert tonic inhibition to suppress the firing of dopamine neurons (*Eshel et al., 2015*).

## Discussion

Here we report that VTA dopamine neurons, identified based either on traditional waveform criteria or through an optogenetically-validated clustering analysis of their response properties to a conditioned cue, exhibited phasic cue-evoked responses that were influenced by inferred value. These responses were observed even though the critical cue had no prior history of direct pairing with any rewarding event. In addition, they were greater than responses to a control cue that was treated similarly and thus had similar levels of salience or novelty or generalized value, all variables that have been proposed to explain phasic activity in other settings that appeared to be at odds with the standard explanation of phasic dopamine activity (*Kakade and Dayan, 2002*; *Bromberg-Martin et al., 2010*; *Matsumoto and Hikosaka, 2009*). These data show that the phasic activity of dopamine neurons can reflect information about value that is not contemplated by TD models, at least as they have been applied to understand the phasic firing of these neurons (*Glimcher, 2011*; *Schultz, 2002*; *Niv and Schoenbaum, 2008*; *Clark et al., 2012*).

Our finding is consistent with a number of recent reports, suggesting that dopamine neurons are likely to access more complex information than is available to standard TD models. For example, dopamine neurons in the rat VTA utilize input from the orbitofrontal cortex to disambiguate states that are not easily distinguished via external information in order to more accurately calculate prediction errors (*Takahashi et al., 2011*). While this result does not require the use of inference in calculating errors, merely access to state information, it suggests that dopamine neurons have access to a major source of this information, given the central role of the orbitofrontal cortex in inference-based behavior (*Stalnaker et al., 2015*).

The phasic activity of dopamine neurons has also been shown to track the value of one cue after changes in the value of an associated cue (*Bromberg-Martin et al., 2010b*). Again these data suggest that dopamine neurons have access to higher order information, which could be described as inference. Indeed these authors describe their results in terms of inference; however, as they note in their discussion (*Bromberg-Martin et al., 2010b*, *final paragraph of discussion*), the inference seen in their task may differ from that shown here in that it does not require access to model-based information, but could instead be based on direct, 'cached' value from earlier training sessions.

Finally elevated dopamine has also been found using microdialysis during an aversive version of the sensory preconditioning task used here (*Young et al., 1998*). However the use of an aversive paradigm, a measurement technique with low temporal resolution, and the lack of control conditions to confirm signaling of cached value errors make it difficult to apply these results to address the very specific proposal that phasic changes in the firing of dopamine neurons signal TD prediction errors in appetitive paradigms.

Our study addresses the limitations of these best available prior reports. We are recording phasic activity of dopamine neurons at their source. We have identified dopamine neurons by two different classification schemes, an old one that has been used repeatedly across labs and species to identify error-signaling dopamine neurons (*Mirenowicz and Schultz, 1994*; *Waelti et al., 2001*; *Tobler et al., 2003*; *Pan et al., 2005*; *Roesch et al., 2007*; *Jo et al., 2013*; *Morris et al., 2006*; *Jin and Costa, 2010*), as well as a new, optogenetically-validated approach that has identified error signals in mice (*Cohen et al., 2012*) and is favored by those that dislike the use of waveform criteria (*Margolis et al., 2006*). We used a carefully designed behavioral preparation in which our critical cue of interest has no prior history of association with reward, thus unique firing to this cue cannot be explained as any sort of cached value (see '*A comment on the basis of responding to the preconditioned cue*' in the behavioral Methods for further information). Further this appetitive task includes two important control cues: one designed to rule out explanations based on generalization and salience (cue C) and another designed to reveal cached value prediction errors (cue B). The inclusion of this cue in particular is important because it allows us to assess the relationship between traditional error signals and any influence of inferred value on the firing of the dopamine neurons.

The close relationship in the firing of the dopamine neurons to B, the cue directly paired with reward, and A, the cue that predicts reward only through B, suggests that whatever is ultimately signaled when a cue with inferred, model based value is presented may be similar to what the same neurons signal in response to the unexpected appearance of a cue that has been directly paired with reward. While this might be explained as error signaling to B, calculated from TD models, and error signaling to A, calculated from something beyond TD models, this solution is cumbersome,

particularly given that inferred and experienced value are actually confounded for a cue directly paired with reward (a fact illustrated by the normal efficacy of reinforcer devaluation at changing conditioned responding [*Holland and Straub, 1979*]). A more parsimonious explanation is that dopamine neurons, unlike standard TD models, have access to a wide variety of information when computing expected value. And that while their firing may conform to what is expected for errors calculated from TD models in some special cases, they may not be signaling TD derived errors. Such a suggestion aligns nicely with recent proposals that dopamine neurons signal errors based on changes in economic utility (*Schultz et al., 2015*), and it would be consistent with data presented in abstract form suggesting that cue-evoked dopamine release in nucleus accumbens is sensitive to devaluation of a paired reward (*Martinez et al., 2008*), though it contradicts data only just published from a similar study in which cue-evoked release was not immediately altered when reward value was manipulated via salt depletion (*Cone et al., 2016*). This variability in correspondence between our unit data and evidence from studies of dopamine efflux in accumbens may reflect to the different dynamics of the two processes or it may indicate some specificity with regard to the information content of the dopaminergic afferents in accumbens versus other areas.

Finally it is important to explicitly note that the general proposal that phasic changes in dopamine are a TD error signal incorporates two very separate sets of predictions. One set, most relevant to the single unit correlates that form the basis of this hypothesis, concerns the information used to construct the error signals. This is obviously the part of the question we have addressed in the current study. That is, do dopaminergic errors reflect only model-free information derived from TD systems or do they also incorporate the predictions of non-TD, model-based systems? We believe our data favor the latter position.

The second set of predictions, not addressed by our study, concerns what the dopaminergic errors do downstream. Do they act only to stamp in the so-called cached values that are acquired through learning in TD models or do they act more broadly to facilitate increases in the strength of associative representations in a way that is orthogonal to distinctions between the systems, model-free or model based, in which those representations reside? The latter role would be more in accord with earlier learning theory accounts that viewed prediction errors as acting on the strength of associative representations (*Glimcher, 2011*; *Bush and Mosteller, 1951*; *Rescorla et al., 1972*). Importantly the answer to the second question is formally separate from the answer to the first. In other words, phasic changes in dopamine may reflect model-based information and yet only act to support model-free, cached-value learning. Or phasic changes in dopamine could act more broadly, supporting both model-free and model-based learning, even if they only reflected value predictions from the former system.

Notably the jury remains out on what sort of learning the brief phasic changes in dopamine thought to signal prediction errors serve to support. In support of dopamine's role in supporting model-based learning, prediction errors observed in ventral striatal target regions seem to reflect both model based and model free information (*Daw et al., 2011*). Further studies have shown that elevated dopamine levels, either observed (*Deserno et al., 2015*) or directly manipulated (*Wunderlich et al., 2012*; *Sharp et al., 2016*), bias subjects towards making model-based decisions, as do changes to dopaminergic gene expression (*Doll et al., 2016*). While suggestive, these studies do not directly distinguish the effects of phasic changes in dopamine neuron firing or release from the effects of slower tonic changes. Such tonic changes may play a very different role from the phasic error signals observed in single unit activity (*Hamid et al., 2016*; *Niv et al., 2007*). Further these studies do not isolate the effects of errors themselves, independent from other confounding variables. Such isolation and specificity can be achieved using Cre-driver lines in rodent species, and as noted earlier, there is now strong evidence from such studies that brief, phasic changes in dopamine neuron firing can act like a prediction error, at least in some downstream targets and behavioral paradigms (*Steinberg et al., 2013*; *Chang et al., 2016*). However the behavior supported by the artificially induced prediction errors in these experiments may be either model-based or model-free (or a mixture). A more definitive answer to this question will require these approaches to be married to paradigms that distinguish these two types of learning.

# Materials and methods

## Subjects

14 adult Long-Evans rats (10 male, 4 female weighing 275–325 g on arrival) were individually housed and given ad libitum access to food and water, except during behavioral training and testing, which which they received 15 min of ad-lib water access following each training session. Rats were maintained on a 12-hr light/dark cycle and trained and tested during the light cycle. Experiments were performed at the National Institute on Drug Abuse Intramural Research Program, in accordance with NIH guidelines. The number of subjects was chosen to have sufficient power to assess learning on the final test-day (*Jones et al., 2012*), and to gather a sufficient number of isolated neurons (>100) for subsequent analysis on the final test day.

## Apparatus

Behavioral training and testing were conducted in standard behavior boxes with commercially-available equipment (Coulbourn Instruments, Allentown, PA). A recessed dipper was placed in the center of the right wall approximately 2 cm above the floor. The dipper was mounted outside the behavior chamber and delivered 40 ul of flavored milk (Nestle) per dipper elevation. Auditory cues (tone, siren, 2 Hz clicker, white noise) calibrated to ~65 dB were used during the behavioral testing.

## Surgical procedures

Rats underwent surgery for implantation of chronic recording electrode arrays. Rats were anesthetized with isoflurane and placed in a standard stereotaxic device. The scalp was excised, and holes were bored in the skull for the insertion of ground screws and electrodes. Multi-electrode bundles [16 nichrome microwires attached to a microdrive] were inserted 0.5 mm above dorsal VTA [anteroposterior (AP) 5.4 mm and mediolateral (ML) 0.8 mm relative to bregma (Paxinos and Watson, 1998); and dorsoventral (DV) 7.0 mm from dura]. In 3 rats, microwire electrodes were also implanted 0.5 mm above ipsilateral orbitofrontal cortex [AP 3.2 mm and ML 3.0 mm relative to bregma (Paxinos and Watson, 1998); and DV 4.0 mm from the dura], and in 2 other rats, microwire electrodes were also implanted 0.5 mm above ipsilateral ventral striatum [AP 1.0 mm and ML 3.0 mm relative to bregma (Paxinos and Watson, 1998); and DV 6.0 mm from the dura]. Once in place, the assemblies were cemented to the skull using dental acrylic. Six rats also received infusions of 1.0 ul of AAV5-DIO-HMD4 into central VTA [anteroposterior (AP) 5.4 mm and mediolateral (ML) 0.8 mm relative to bregma, and 8.1 mm below dura]; there were no effects of this treatment on any of the results we have reported.

## Behavioral training

Rats began sensory preconditioning 2 weeks after electrode implantation. The sensory preconditioning procedure consisted of three phases, of similar design to a prior study (*Jones et al., 2012*).

### Preconditioning

Rats were shaped to retrieve a liquid reward from a fluid dipper over three sessions; each session consisted of twenty deliveries of 40 ul of flavored milk. After this shaping, rats underwent 2 days of preconditioning. Each day of preconditioning, rats received twelve trials in which two pairs of auditory cues (A->B and C->D) were presented sequentially, with no delay between cues, six times each, in a blocked design. Cues were each 10 s long, the inter-trial intervals varied from 3 to 6 min, and the order the blocks alternated across days. Cues A and C were white noise or clicker (counterbalanced), and cues B and D were siren or tone, (counterbalanced).

### Conditioning

After preconditioning, rats underwent conditioning. Each day, rats received a single training session, consisting of six trials of cue B paired with the flavored milk reward and six trials of D paired with no reward. The flavored milk reward was presented three times via the dipper in the food cup at 1, 4, and 7 s into the 10 s presentation of cue B. Cue D was presented for 10 s without reward. The two cues were presented in 3-trial blocks, counterbalanced. The inter-trial intervals varied between 3 and 6 min. Ten rats were given 6 days of conditioning, while two were advanced to the probe test after

5 days due the presence of putative wide waveform neurons at the beginning of the sixth conditioning day. There was no difference between these groups in the final test.

## Probe test

After conditioning, the rats underwent a single probe test, which consisted of three reminder trials of B paired with reward and three trials of D unpaired interleaved, followed by presentation of cues A and C, alone, six times each, without reward, with the presentation order counterbalanced across animals. In six animals A and C trials were blocked, while in 8 animals they were interleaved; both groups showed all reported effects and were merged. Cue durations, timing of reward, and inter-trial intervals were as above.

## A comment on the basis of responding to the preconditioned cue

We have interpreted our sensory preconditioning effect in terms of an associative chaining or value inference mechanism. An alternative account, which has been employed in other recent studies using similar procedures (*Kurth-Nelson et al., 2015*; *Wimmer and Shohamy, 2012*), is that the conditioned responding to cue A results from mediated learning that occurs during the second phase of the experimental procedure (*Rescorla and Freberg, 1978*). Briefly, this account suggests that following the initial pairings of A and B, subsequent presentations of B for conditioning activate a representation of A in memory within a relatively close temporal contiguity with the delivery of sucrose, resulting in the representation of A becoming directly associated with this reward. If this were to occur, then at test, the subsequent conditioned responding to A might reflect the cue's direct association with sucrose, rather than requiring B to bridge the experiences of A and sucrose.

While there is significant evidence within the literature for the phenomenon of mediated learning (reviewed in *Ward-Robinson and Hall, 1996*; *Holland, 1990*), several features of our behavioral design were chosen to bias strongly against the operation of this mechanism.

First, we used forward (A→B) rather than simultaneous (AB) or backward pairings (B→A) of the pre-conditioned and conditioned cues. This is important because mediated learning in rodents has been suggested to operate primarily when A and B are presented simultaneously (*Rescorla and Freberg, 1978*) or as the serial compound B→A (i.e. backward sensory preconditioning; 52). The reason for this is intuitive because either of these temporal arrangements maximizes the chances that B will evoke a representation of A during the conditioning phase and concurrent with reward delivery, an arrangement that obvious benefits in maximizing the ability of an evoked representation of A to become directly associated with reward.

Our design avoids this by initially presenting A and B as the serial compound A→B, an arrangement which, as far as B is concerned, parallels a backward conditioning procedure. Backward cue-outcome pairings have generally been shown to yield weak, if any, excitatory conditioning (*Mackintosh, 1974*). For this reason, it is widely accepted that A→B pairings will render B relatively ineffective at subsequently conjuring up a memory of A, thus making the contribution of mediated learning insubstantial (*Hall, 1996*).

Second, the amount of training given in Phase 2 of conditioning, with B-reward pairings, was also designed to discourage mediated learning. As noted above, the presentation of B in conditioning should activate a representation of A in memory. However, with repeated presentations of B without A, the representation of A evoked by B will also extinguish. For this purpose, we present 3 times as many B (not A) trials in the conditioning phase as we present A and B pairings, further undermining the likelihood that an evoked A representation will be maintained.

In conclusion, we believe our implementation of these specific behavioral parameters should largely eliminate any potential contribution of mediated learning to the sensory preconditioning effect in our particular design, and favor the parsimonious interpretation of the sensory preconditioning effect in terms of an associative chaining or value inference mechanism. We would note that this interpretation is supported by our own prior report that OFC inactivation at probe test in this exact paradigm abolishes responding to A and has no effect on responding to B (*Jones et al., 2012*), since mediated learning is basically simple conditioning and OFC manipulations typically have no effect on expression of previously acquired conditioned responding (*Takahashi et al., 2009*; *2013*).

## Electrophysiology

Neural signals were collected from the VTA during each behavioral session. Differential recordings were fed into a parallel processor capable of digitizing 16-to-32 signals at 40 kHz simultaneously (Plexon). Discriminable action potentials of <3:1 signal/noise ratio were isolated on-line from each signal using an amplitude criterion in cooperation with a template algorithm. Discriminations were checked continuously throughout each session. Time-stamped records of stimulus onset and neuronal spikes were saved digitally, as were all sampled spike waveforms and the discrimination file. Off-line re-analysis incorporating 3D cluster-cutting techniques confirmed and corrected on-line discriminations. Except where explicitly noted, all neurons identified via off-line sorting were included in each analysis.

## Statistical analyses

Raw data were processed with Matlab to extract food cup entries and spike-timing relative to cue-onset. Entries were converted to a response measure: the percentage of time rats spent with their head in the food cup during cue presentation as measured by an infrared photo beam positioned at the front of the food cup. Spike times were binned and analyzed as specified below. In comparing cue-evoked to reward-evoked activity, bins spanning the first 500ms of each period were analyzed. In comparing response differences evoked by different cues, bins spanning the first 1 s of cue-evoked activity were analyzed. For all statistical tests, an alpha level of 0.05 was used.

## AUC calculation

As per prior reports (*Cohen et al., 2012*), we normalized the firing rate of individual neurons by comparing the histogram of spike counts during each bin of spiking activity (100 ms, test bins from each trial for a cue, at a particular time post-stimulus) against a histogram of baseline (100 ms) bins, from all trials for that cue. The ROC in question is calculated by normalizing all test and baseline bin counts, such that the minimum bin count was 0 and the maximal bin count was 1 sliding a discrimination threshold across each histogram of bins, from 0 to 1 in 0.01 steps, such that fraction of test bins identified above the threshold was a 'true positive' rate and the fraction of baseline bins above the threshold was a 'false negative' rate for an ROC curve. The area under this curve was then estimated by trapezoidal numerical estimation, with an auROC below 0.5 being indicative of inhibition, and an auROC above 0.5 being indicative of excitation above baseline.

## Classification of dopamine neurons by response dynamics

In order to isolate VTA neuron response types shown to be indicative of putative dopaminergic and GABAergic genetic identities (*Cohen et al., 2012*), we took the auROC normalized responses of neurons during their response to the cue predictive of reward (cue B), and performed a simple classification to separate neural responses. We first performed principal components analysis on a matrix of neural responses during cue B and reward presentation (neuron-by-time) to simplify the neural dynamics to the 3 most descriptive ways in which neurons differed. We then classified this description of the neural population (first 3 principal components) with a simple unsupervised hierarchical clustering algorithm, finding the similarity (Euclidean distance) between all pairs of neurons in principal components space, and iteratively grouping the neurons them into larger and larger clusters on the basis of their similarity (i.e. agglomerative complete-linkage clustering). A distance-criterion was then set to extract exactly 3 clusters from this hierarchical tree.

## Classification of dopamine neurons by waveform

Neurons were screened for wide waveform and amplitude characteristics, calculated on their mean action-potential across a recording session. Neurons were identified as dopaminergic if their negative half-width exceeded a standard criterion (450 μs) and the ratio of (max) positive to (min) negative voltage deflections was greater than zero (*Mirenowicz and Schultz, 1994*; *Takahashi et al., 2011*; *Roesch et al., 2007*; *Jo et al., 2013*). Four such neurons were identified as wide waveform across the probe sessions.

### Histology

After the final recording session, rats were euthanized and perfused first with PBS and then 4% formalin in PBS. Electrolytic lesions (1 mA for 10 s) made just before perfusion were examined in fixed, 0.05 mm coronal slices stained with cresyl violet. Anatomical localization for each recording was verified on the basis of histology, stereotaxic coordinates of initial positioning, and recording notes.

## Acknowledgements

This work was supported by the Intramural Research Program at the National Institute on Drug Abuse. The opinions expressed in this article are the authors' own and do not reflect the view of the NIH/DHHS.

## Additional information

### Funding

| Funder | Grant reference number | Author |
| --- | --- | --- |
| National Institute on Drug Abuse | IRP | Geoffrey Schoenbaum |

The funders had no role in study design, data collection and interpretation, or the decision to submit the work for publication.

### Author contributions

BFS, Conception and design, Acquisition of data, Analysis and interpretation of data, Drafting or revising the article; JLJ, GS, Conception and design, Analysis and interpretation of data, Drafting or revising the article

### Ethics

Animal experimentation: Experiments were performed at the National Institute on Drug Abuse Intramural Research Program in accordance with NIH guidelines and an approved institutional animal care and use committee protocol (15-CNRB-108). The protocol was approved by the ACUC at NIDA-IRP (Assurance Number: A4149-01).

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
