## [Decision Letter]

Thank you for submitting your work entitled "VTA dopamine neurons compute inferred and cached value (TD) prediction errors in a common framework" for consideration by *eLife*. Your article has been reviewed by three peer reviewers, and the evaluation has been overseen Timothy Behrens as the Senior and Reviewing Editor. The following individuals involved in review of your submission have agreed to reveal their identity: Nathaniel Daw (Peer reviewer) and Paul Phillips (Peer reviewer).

The reviewers have discussed the reviews with one another and the Reviewing Editor has drafted this decision to help you prepare a revised submission.

Summary:

Sadacca and colleagues carried out a sensory preconditioning task to test whether dopamine transmission encodes inferred values of sensory stimuli that had not been directly paired to rewards. They report that when a cue is paired with reward, another cue associated with it (through sensory preconditioning) acquires a dopamine response to its presentation even though the latter cue has never been directly paired with reward. This type of association is not available from standard model-free reinforcement learning algorithms. Therefore, the authors conclude that dopamine transmission can compute prediction errors through (model-based) inference.

All three reviewers regarded this as a potentially major and important result that should be celebrated. For example:

This wonderful study recorded the activity of midbrain dopamine neurons during sensory preconditioning. This is an absolutely critical, long-overdue test of a fundamental question about dopaminergic 'prediction error' signals: exactly what 'prediction' do they use to compute their 'error'? I've been pestering rodent researchers to do this experiment for years (this type of experiment requires naïve subjects and hence is much more easily done in rodents than in monkeys).

This article demonstrates that (putative) dopamine neurons in rat VTA compute reward prediction errors with respect to reward predictions derived by combining information from separate sensory preconditioning and reward conditioning phases in a manner that the simplest TD learning algorithms cannot. This is an important and extremely clean result.

This work is potentially very important and the manuscript is clearly written.

Essential revisions:

However, all three reviewers highlighted several major concerns that need to be dealt with. The essential concerns are in fact very similar between the three reviewers. I have left the native reviews below because each reviewer raises very similar points, but there is nuance in the different expressions of the concerns that, I think, will be useful in preparing the revision.

The essential concerns can be divided, I think, into three categories. Clarification of procedures and results, appropriate reflection of the existing literature, and some extra information (or discussion) about some slightly concerning features of the data – in particular the response to stimulus D.

In the discussion between reviewers, it was clear that the first two of these categories were essential. With respect to last category, there were some discussions about potential causes that might both mitigate some of the concerns below, and suggest some alternative tests. For example,

"My hunch is that D gained a positive response due to the sort of 'generalization'/'alerting'/'pseudo-conditioning' phenomenon that's often seen where if two stimuli of the same modality are presented in the same task environment, they both gain some positive short-latency DA response even if only one is paired with rewards (e.g. old Schultz studies, recent study by Kobayashi and Schultz, voltammetry study by Day et al. 2007, etc.). This effect may be weaker in A and C because during their phase of training session no rewards were delivered."

"A key question here is whether there is bias in the probe phase due to selecting cells based on their response to B and testing them on the same data. I think we all agree that this doesn't bias the key result (the comparison between A and C in Figure 3) but I think it might indeed bias Figure 3 (which is based in part on the responses to B and D) and more obviously the extended versions of Figure 3 that I think at least two of us asked for (i.e. comparing a scatterplot of that sort between neuronal subtypes). Could they do analyses based on holdouts to avoid directly using the same trials to select and test?"

*Reviewer #1:*This wonderful study recorded the activity of midbrain dopamine neurons during sensory preconditioning. This is an absolutely critical, long-overdue test of a fundamental question about dopaminergic 'prediction error' signals: exactly what 'prediction' do they use to compute their 'error'? I've been pestering rodent researchers to do this experiment for years (this type of experiment requires naïve subjects and hence is much more easily done in rodents than in monkeys). When I saw this data presented at a recent SfN I was very pleased and eager to see it get in print, so I'm happy to see it here at last!

I like the very clean experimental design that allows measurement and direct comparison between conventional signals consistent with TD learning models as the new unconventional signals. I also like that the authors cite previous work very well, even citing a related voltammetry study from the Phillips group that has only appeared in abstract form. I also appreciate that the paper is a short, sweet, and to the point test of its main hypothesis.

In fact, the paper may have gone slightly too far in this direction. My major comments are about the more sophisticated interpretations and subtler findings that the authors seem to have left out of their short and sweet narrative.

1) "After conditioning, the rats underwent a single probe test, which consisted of three reminder trials of B paired with reward and three trials of D unpaired, followed by presentation of cues A and C, alone, six times each, without reward." This needs to be clarified. Were the B and D reminder trials done in a fixed sequence (e.g. B, B, B, D, D, D), were they randomly interleaved, or something else? The same needs to be clarified for the A and C probe presentations.

This small detail is critical to the logic of the paper. The cues should ideally have been presented in randomized orders. If subjects were always probed with A before C then conceivably the greater behavioral response to A than C could have been due to an effect of time rather than due to a discrimination between the meanings of the cues. For instance, suppose something about the reminder treatment (e.g. the presentations of B/D or the deliveries of rewards) put the animals in a state of generalized enhanced responsiveness to the A/C cues, a state which slowly waned with time or habituated with repeated presentation of A/C cues. If A was always presented before C, then A would have produced greater behavioral responses than C, even if animals failed to assign higher inferred value to A or even if they failed to discriminate at all between A and C.

2) Figure 3 has straightforward logic, but two key points need to be clarified:

2.1) "Notably this was also true for a handful of neurons that exhibited the classic wide, polyphasic waveforms traditionally used to identify dopamine neurons (Figure 3, filled circles)." It's very hard to make out the data points for the three types of neurons. How many neurons is in the handful (n)? Was the effect significant? The authors present clear statistics for the Type 2 cells but need to present the same statistics for the classically defined DA neurons.

2.2) What responses did cues A and C evoke in the type 1 and 3 neurons, and do they have any significant A-C effect? This is of great interest because these neurons were recently prominently proposed to have a major role in computing DA prediction error signals (Eshel et al., Nature 2015), so one would expect that the important new computations these authors have discovered in DA neurons should be presaged by similar computations in type 1 and 3 neurons.

Currently the authors address a partly related but less critical point. They say these neurons don't have a significant correlation between onset firing for B-D and A-C (without presenting statistics to justify this statement), though that's perhaps to be expected given that their responses are tonic rather than phasic.

The above points 2.1 and 2.2 could be cleared up by showing the same main results that are shown for the type 2 cells (activity plots in Figure 3 and the statistical tests of population responses to B-D and A-C), also for the classical electrophysiologically defined putative DA neurons, and also for the type 1 and type 3 cells. This will also help validate their classification by confirming that the putative DA, type 1 cells, and type 3 cells behaved consistently with prior optogenetically-verified studies.

3) The authors chose the word "inference" as the key word for the novel part of their study, drawing a distinction between "inferred value" and "cached value".

Why not use "model-based"? This is the term that was prominently defined and contrasted with the "cached value" that motivates this study, in a seminal paper on this topic (Daw et al., Nature Neuroscience 2005). It's surprising that the authors don't cite this paper, since it seems to be the origin for the "cached values" terminology the authors use throughout their study and in their title. I find it strange that the authors don't discuss or even mention the concept of model-based vs model-free learning. In that terminology, this study is very important: it is the first, long-awaited test of popular hypotheses about whether DA neurons use model-based or cached values!

"Inference" is a very broadly-defined word that has different meanings in different contexts. If the authors want to use it I would appreciate it if they defined which specific form of inference they're studying, rather than treating "inference" as universally synonymous with model-based learning. Notably this paper places "inference" in contrast with TD learning, but in that field of machine learning "inference" has a more general meaning. There are forms of TD learning that learn without model-based reasoning but which still clearly use forms of inference (e.g. Bayesian Q-learning, which uses Bayesian inference (Dearden, Friedman, and Russell, AAAI 1998)).

Furthermore, "inference" has been used in previous work on dopamine neurons and TD errors, but the present work uses it in a different manner without clearly explaining the difference in definitions. The cited paper by Bromberg-Martin et al. (2010) defined "inferred stimulus value" as updating the value of a stimulus without experiencing that stimulus. This paper, however, uses "inferred value" in a different way apparently synonymous with model-based value. This un-discussed difference in terminology makes it sound like the authors are (perhaps unintentionally) dismissing the previous work as invalid. The authors seem to state that study was merely "suggestive" of inferred prediction errors because it was subject to "confounding" issues, but it would be more accurate to say that both studies had valid results and were simply studying different forms of inference, or were defining "inference" in different ways.

Reviewer #2:

This article demonstrates that (putative) dopamine neurons in rat VTA compute reward prediction errors with respect to reward predictions derived by combining information from separate sensory preconditioning and reward conditioning phases in a manner that the simplest TD learning algorithms cannot. This is an important and extremely clean result, but of interest in great part because it plays into a large literature on this topic in computational modeling and human neuroimaging which is entirely and surprisingly neglected here but provides context and subtlety to the interpretation.

1) My main concern is sort of semantic, but both the Abstract and the final sentence of the article claim that these signals are not TD errors. In my view, a "TD error" is a Bellman residual, the temporal difference between the reward predicted at times t vs t+1, which is precisely consistent with these responses. The issue is where the predictions themselves come from, i.e. whether they can have themselves been learned by caching simple adjustments driven by previous TD errors, vs. some more complicated inference (a computational distinction due to Daw et al., 2005). TD *errors*, in short, need not only arise (only) from TD *learning*, even though the former drive the latter. This was perhaps first pointed out in the context of a human imaging study (Daw et al. 2011) which somewhat presaged these results and their interpretation.

2) A related interpretational subtlety is how the learning and integrative inference that underlies these predictions might have taken place. Again, relevant human imaging work (Wimmer et al., 2012; Kurth-Nelson et al. 2015; Shohamy & Daw 2015) is not discussed; these results suggest that associative retrieval (from B to A) occurs during the conditioning phase, together with which standard TD updating (from the reward to the reactivated A) would suffice. (An alternative – suggested in more human work by Gershman et al., 2012, and computationally by Sutton's, 1991, DYNA – is that replay between the conditioning and transfer phases, driving regular TD updating, would integrate the information.) The former possibility directly instantiates the suggestion, dismissed briefly in the discussion of Bromberg-Martin's (2010) similar result, that this result might be due to simple TD plus some altered state representation (here that state B and A are combined; in general – Dayan 1991 – that states are represented by their successors). I agree that this study is a cleaner and more stripped down test of integrative prediction than Bromberg-Martin's serial reversal, but it is just wrong to distinguish them in the way described here.

Reviewer #3:

Sadacca and colleagues carried out a sensory preconditioning task to test whether dopamine transmission encodes inferred values of sensory stimuli that had not been directly paired to rewards. They report that when a cue is paired with reward, another cue associated with it (through sensory preconditioning) acquires a dopamine response to its presentation even though the latter cue has never been directly paired with reward. This type of association is not available from standard model-free reinforcement learning algorithms. Therefore, the authors conclude that dopamine transmission can compute prediction errors through (model-based) inference.

This work is potentially very important and the manuscript is clearly written. However, one aspect of the control seems to be problematic, making the conclusions of the work somewhat equivocal in my opinion.

Major concern:

The response of putative dopamine neurons to control stimulus D, which was never paired with reward, is quite large. This response was significantly smaller than that for stimulus B in the analysis epoch (first second of cue presentation). However, it is possible that this difference could relate to features other than the reward contingency. Selection criteria for putative dopamine neurons include the responsiveness to a reward-associated cue, specifically stimulus B and so, by design, there is a systematic selection bias towards neurons that respond to stimulus B. Therefore, it is feasible that the increased firing to stimulus B over stimulus D is due to the sensory properties of stimulus B since regardless of which auditory stimulus was designated as stimulus B in a particular subject, neurons were selected that responded to that stimulus. The authors need to address this potential confound, especially given that the mode response of the putative dopamine neurons did not discriminate between stimuli B and D (Figure 3).

Stimulus A and C also produced significant increases in the firing of putative dopamine neurons. The key comparison of the study was between these two responses, which was indeed, significantly greater for stimulus A. However, it is disconcerting how these responses spanned the response to stimulus D since the B to D and A to C comparisons appear (although it's not completely clear from the manuscript) to have been carried out independently without consideration of the variance of the responses to all of the stimuli. The concern is that if all the stimuli that were not directly paired with reward (A, C and D) were compared, significance would be lost. The authors should test whether the differences between responses to A and C are still significant if the analysis includes the responses to all of the stimuli (two-way ANOVA or comprehensive one-way ANOVA). At very least the authors should discuss if and why the responses to stimuli C and D are not more similar (e.g., more generalization to reward).

---

## [Author Response]

*Essential revisions:However, all three reviewers highlighted several major concerns that need to be dealt with. The essential concerns are in fact very similar between the three reviewers. I have left the native reviews below because each reviewer raises very similar points, but there is nuance in the different expressions of the concerns that, I think, will be useful in preparing the revision.The essential concerns can be divided, I think, into three categories. Clarification of procedures and results, appropriate reflection of the existing literature, and some extra information (or discussion) about some slightly concerning features of the data* – *in particular the response to stimulus D.*

In the discussion between reviewers, it was clear that the first two of these categories were essential. With respect to last category, there were some discussions about potential causes that might both mitigate some of the concerns below, and suggest some alternative tests. For example, "My hunch is that D gained a positive response due to the sort of 'generalization'/'alerting'/'pseudo-conditioning' phenomenon that's often seen where if two stimuli of the same modality are presented in the same task environment, they both gain some positive short-latency DA response even if only one is paired with rewards (e.g. old Schultz studies, recent study by Kobayashi and Schultz, voltammetry study by Day et al. 2007, etc.). This effect may be weaker in A and C because during their phase of training session no rewards were delivered.""A key question here is whether there is bias in the probe phase due to selecting cells based on their response to B and testing them on the same data. I think we all agree that this doesn't bias the key result (the comparison between A and C in Figure 3) but I think it might indeed bias Figure 3 (which is based in part on the responses to B and D) and more obviously the extended versions of Figure 3 that I think at least two of us asked for (i.e. comparing a scatterplot of that sort between neuronal subtypes). Could they do analyses based on holdouts to avoid directly using the same trials to select and test?"

We thank the reviewers for their thoughtful and thorough analysis of our manuscript. As noted by the reviewers, the initial manuscript was “too short and sweet”; in our earnestness to make a concise, accessible report of our findings, we neglected discussion of some features of the data and several findings in the literature, and are grateful for the opportunity to expand our Discussion. In the included revision, we’ve clarified the procedures used in our behavior task, extended our discussion of the relevant literature and reanalyzed our data as requested.

Reviewer #1:1) "After conditioning, the rats underwent a single probe test, which consisted of three reminder trials of B paired with reward and three trials of D unpaired, followed by presentation of cues A and C, alone, six times each, without reward." This needs to be clarified. Were the B and D reminder trials done in a fixed sequence (e.g. B, B, B, D, D, D), were they randomly interleaved, or something else? The same needs to be clarified for the A and C probe presentations.This small detail is critical to the logic of the paper. The cues should ideally have been presented in randomized orders. If subjects were always probed with A before C then conceivably the greater behavioral response to A than C could have been due to an effect of time rather than due to a discrimination between the meanings of the cues. For instance, suppose something about the reminder treatment (e.g. the presentations of B/D or the deliveries of rewards) put the animals in a state of generalized enhanced responsiveness to the A/C cues, a state which slowly waned with time or habituated with repeated presentation of A/C cues. If A was always presented before C, then A would have produced greater behavioral responses than C, even if animals failed to assign higher inferred value to A or even if they failed to discriminate at all between A and C.

We agree that presenting exclusively one cue followed by the other in the final test could skew neural and behavioral responses to the cues. In order to control for this feature in this experiment, A/C presentation order was counterbalanced between rats (in half of our animals, A was presented first, and half C came first). A direct comparison of data from the differently-balanced sub-groups showed that this ordering was not responsible for the greater neural activity observed to A vs C: putative dopamine neurons recorded in rats presented with C first still showed significantly more firing to A than C (t=4.6, df = 29, p<0.01), and the difference was similar that in the rats that received A first (t = 3.2, df = 21, p<0.01), as were the B/D vs A/C correlations (C first: r = 0.67, p<0.01; A first: r=0.70, p<0.01).

Presentation sequence was also of interest: the first six animals run proceeded as previously (1, 2) in blocks of 6 trials of A or C, followed by 6 trials of the other cue, while the final 8 animals had these 12 trials interleaved A, C, A, C or C, A, C, A; the effects (A/C response difference, A/C response vs B/D response correlation) were similar in each group, so the two groups were pooled. In addition, to clarify, on the final probe day, B/D presentation was interleaved (B, D, B, D) to avoid a string of rewarded or non-rewarded trials prior to the critical preconditioned cues.

The reason for this is that we were essentially moving from the task that we had used previously while optimizing for recording. Since the changes did not seem to affect the findings, we ultimately used all the data. But we agree these details are important because of the potential confounds, so in the revised manuscript we describe the counterbalancing and sequence in probe trials in more detail in the Methods, where we state: “In six animals A and C trials were blocked, while in 8 animals they were interleaved; both groups showed all reported effects and were merged. “

2) Figure 3 has straightforward logic, but two key points need to be clarified:2.1) "Notably this was also true for a handful of neurons that exhibited the classic wide, polyphasic waveforms traditionally used to identify dopamine neurons (Figure 3, filled circles)." It's very hard to make out the data points for the three types of neurons. How many neurons is in the handful (n)? Was the effect significant? The authors present clear statistics for the Type 2 cells but need to present the same statistics for the classically defined DA neurons.

There were 4 neurons with ‘wide waveforms’ 1, one classified as tonically excited and three classified as phasically excited; while the direction of each of these responses matched our predictions (mean response for B>D and A>C) and therefore provide some support that our primary measure of neuron selection is pulling out reasonable features of the data, our population of wide waveform neurons is not large enough to stand alone (i.e. to assess group significance for B/D responding, A/C responding or correlations between the measures). Indeed it would be a near impossible task to record enough of these neurons in this sort of “single-shot” study. However we think it is important to include them for completeness and to help relate the rodent data to classic findings. And we are happy to provide as much information as possible. To this end, we have added supplemental Figure 1 showing example unit responses for wide-waveform neurons to give the reader an intuition of the nature of these responses.

2.2) What responses did cues A and C evoke in the type 1 and 3 neurons, and do they have any significant A-C effect? This is of great interest because these neurons were recently prominently proposed to have a major role in computing DA prediction error signals (Eshel et al., Nature 2015), so one would expect that the important new computations these authors have discovered in DA neurons should be presaged by similar computations in type 1 and 3 neurons.Currently the authors address a partly related but less critical point. They say these neurons don't have a significant correlation between onset firing for B-D and A-C (without presenting statistics to justify this statement), though that's perhaps to be expected given that their responses are tonic rather than phasic.The above points 2.1 and 2.2 could be cleared up by showing the same main results that are shown for the type 2 cells (activity plots in Figure 3 and the statistical tests of population responses to B-D and A-C), also for the classical electrophysiologically defined putative DA neurons, and also for the type 1 and type 3 cells. This will also help validate their classification by confirming that the putative DA, type 1 cells, and type 3 cells behaved consistently with prior optogenetically-verified studies.

Cues A and C had an effect on the responding of tonically excited neurons and we now include analysis of tonically activated neurons in panels supplementing Figure 3 – 3.2 and 3.3. Tonically activated neurons discriminated A/C reliably, while tonically inhibited neurons reliably discriminated neither B/D nor A/C during the first second of cue response. We discuss this activity at the manuscript at the end of the Results section. In addition, we did observe a positive correlation between A-C/B-D for both groups of tonically activated neurons but this did not reach significance for either type as classified here.

3) The authors chose the word "inference" as the key word for the novel part of their study, drawing a distinction between "inferred value" and "cached value".Why not use "model-based"? This is the term that was prominently defined and contrasted with the "cached value" that motivates this study, in a seminal paper on this topic (Daw et al., Nature Neuroscience 2005). It's surprising that the authors don't cite this paper, since it seems to be the origin for the "cached values" terminology the authors use throughout their study and in their title. I find it strange that the authors don't discuss or even mention the concept of model-based vs model-free learning. In that terminology, this study is very important: it is the first, long-awaited test of popular hypotheses about whether DA neurons use model-based or cached values!"Inference" is a very broadly-defined word that has different meanings in different contexts. If the authors want to use it I would appreciate it if they defined which specific form of inference they're studying, rather than treating "inference" as universally synonymous with model-based learning. Notably this paper places "inference" in contrast with TD learning, but in that field of machine learning "inference" has a more general meaning. There are forms of TD learning that learn without model-based reasoning but which still clearly use forms of inference (e.g. Bayesian Q-learning, which uses Bayesian inference (Dearden, Friedman, and Russell, AAAI 1998)).

We apologize that we neglected to cite Daw’s 2005 paper, as this paper has played a prominent role in our thinking on these models of decision-making. It and several others are now cited. Our use of the terminology inference vs. cached was simply because initial readers found the ‘psychological’ terminology more accessible, and we were concerned a proper treatment of the model-based vs. model free framework wouldn’t fit in a shorter manuscript. In the revised manuscript, we include the terminology of the model-based / model-free framework in the same place we mention of cached or inferred values, in both the Introduction and the Discussion. There we also define more explicitly what we mean by inference. Basically – as the reviewers note – we are using it to mean the same thing as model-based. We now say this clearly in the Introduction.

Actually just for fun, we looked at the definition and etiology of these words. Infer is to deduce or conclude (information) from evidence and reasoning rather than from explicit statements. In the context of a behavior, this sounds very much like what is meant by using a model to arrive at a conclusion as opposed to explicit or direct experience. And the definition of deduce is even more interesting. One source defined it as to figure out something new, based on what you already know, and further noted that:

“it is derived from the Latin ducere, meaning ‘to lead’, so a person who deduces something is "leading" their mind from one idea to the next.”

In any event, we have defined what we mean.

Furthermore, "inference" has been used in previous work on dopamine neurons and TD errors, but the present work uses it in a different manner without clearly explaining the difference in definitions. The cited paper by Bromberg-Martin et al. (2010) defined "inferred stimulus value" as updating the value of a stimulus without experiencing that stimulus. This paper, however, uses "inferred value" in a different way apparently synonymous with model-based value. This un-discussed difference in terminology makes it sound like the authors are (perhaps unintentionally) dismissing the previous work as invalid. The authors seem to state that study was merely "suggestive" of inferred prediction errors because it was subject to "confounding" issues, but it would be more accurate to say that both studies had valid results and were simply studying different forms of inference, or were defining "inference" in different ways.

We did not mean to dismiss or in any way denigrate this result. Indeed we brought it up precisely because we think it was terrific and comes closest to what we have done. But – like our own similar work in OFC where we stole this idea (3) – it cannot rule out prior experience as the root source of the predictions. That was the only point we meant to make. Indeed Bromberg makes this point in his paper, and even suggests the experiment we have done, so in our revision we have simply paraphrased him in our Discussion, with the specific quote we refer to being:

“standard model-based learning is different from the inference seen in our task…..model-based algorithms can infer stimulus values in novel situations, even before a stimulus has been directly paired with reward (e.g., after separately observing A -> B and B -> reward, they infer that A -> reward). In contrast, our task required an inference rule to be learned through extensive training in a familiar task environment.”.

Reviewer #2:1) My main concern is sort of semantic, but both the Abstract and the final sentence of the article claim that these signals are not TD errors. In my view, a "TD error" is a Bellman residual, the temporal difference between the reward predicted at times t vs t+1, which is precisely consistent with these responses. The issue is where the predictions themselves come from, i.e. whether they can have themselves been learned by caching simple adjustments driven by previous TD errors, vs. some more complicated inference (a computational distinction due to Daw et al., 2005). TD *errors*, in short, need not only arise (only) from TD *learning*, even though the former drive the latter. This was perhaps first pointed out in the context of a human imaging study (Daw et al. 2011) which somewhat presaged these results and their interpretation.

As mentioned in response to R1, we agree that the distinction between the information reflected in the error and the use of that error is an extremely interesting question. We did not address it in our original manuscript because it seemed somewhat beyond a general audience, particularly as we are not computational neuroscientists ourselves. However as both R1 and R2 want us to address this, we have adjusted our Introduction and Discussion to distinguish between these ideas and to be more precise about what we are showing, namely that the error signaled by these neurons incorporates model-based information. We believe this is contrary to pure TD models of these errors. However we recognize this does not address whether these errors are then acting within a TD learning system downstream, and we have expanded our Discussion to state this clearly. We regret we do not have the deep knowledge of the computational theory that our reviewers obviously have. If we have not cited appropriately, we are happy to incorporate whatever citations should be included and to make any other changes in the interest of fairness and completeness.

2) A related interpretational subtlety is how the learning and integrative inference that underlies these predictions might have taken place. Again, relevant human imaging work (Wimmer et al., 2012; Kurth-Nelson et al. 2015; Shohamy & Daw 2015) is not discussed;

We apologize for not citing this work. We’ve now included these references.

*these results suggest that associative retrieval (from B to A) occurs during the conditioning phase, together with which standard TD updating (from the reward to the reactivated A) would suffice. (An alternative – suggested in more human work by Gershman et al., 2012, and computationally by Sutton's, 1991, DYNA – is that replay between the conditioning and transfer phases, driving regular TD updating, would integrate the information.) The former possibility directly instantiates the suggestion, dismissed briefly in the discussion of Bromberg-Martin's (2010) similar result, that this result might be due to simple TD plus some altered state representation (here that state B and A are combined; in general* – *Dayan 1991* – *that states are represented by their successors).*

We agree this particular issue of “when is cue A updated?” is important. Again we side-stepped it in the interest of simplicity. However as the reviewers bring it up, we now deal with it directly. We cannot comment on how preconditioning may be happening in other labs’ versions of this task. However though it is difficult to completely rule out, we do not believe this is the basis of responding to the preconditioned cues in our version. To make sure, we designed our task to limit the possibility of associative retrieval during conditioning – so called mediated learning. This was done both by forward pre-conditioning the two cues (ie A->B rather that AB in parallel or even worse, B->A) and also by giving several days of conditioning in which B is presented but A is not. Both choices should in theory minimize mediated learning. We believe this conjecture is strongly supported by data showing that inactivation of the OFC in the probe test abolishes responding to the preconditioned cue in this paradigm without affecting responding to the cue paired directly with reward (1). If responding to the preconditioned cue were due to learning – whether mediated or not – in the second phase, we think we would not see effects of OFC inactivation on subsequent responding, since there is no evidence that OFC is necessary for recalling previously acquired independent bits of information like this (5-8). We now include an extensive consideration of this potential alternative interpretation along with our evidence against it in the revised manuscript in a “comment” that we have placed in the behavioral Methods section.

I agree that this study is a cleaner and more stripped down test of integrative prediction than Bromberg-Martin's serial reversal, but it is just wrong to distinguish them in the way described here.

We apologize if we inadvertently and unfairly distinguished our results from this prior report. To correct this, we now simply quote Bromberg’s own discussion of this result, in which he makes much more succinctly the same points we were trying to make.

Reviewer #3:The response of putative dopamine neurons to control stimulus D, which was never paired with reward, is quite large. This response was significantly smaller than that for stimulus B in the analysis epoch (first second of cue presentation). However, it is possible that this difference could relate to features other than the reward contingency. Selection criteria for putative dopamine neurons include the responsiveness to a reward-associated cue, specifically stimulus B and so, by design, there is a systematic selection bias towards neurons that respond to stimulus B. Therefore, it is feasible that the increased firing to stimulus B over stimulus D is due to the sensory properties of stimulus B since regardless of which auditory stimulus was designated as stimulus B in a particular subject, neurons were selected that responded to that stimulus. The authors need to address this potential confound, especially given that the mode response of the putative dopamine neurons did not discriminate between stimuli B and D (Figure 3).

We agree that magnitude of response to control stimulus D was surprising, but we are confident that the ability of putative DA neurons to discriminate B-D has nothing to do with neural selection. To assess this reasonable concern, we re-sorted the probe day data in two ways and reanalyzed it as before. When we performed a more “B” agnostic version of our initial classification, classifying PSTHs on the basis of the mean activity across all cues (as opposed to just cue B), we recapitulate the categories presented in the manuscript (phasic excitation, tonic excitation, tonic inhibition), and find the same features of those phasic neurons (B/D discrimination, t = 3.7, p> 0.001; A/C discrimination, t = 4.1 p<0.001; B-D & A-C correlation r =0.61, p<0.001). Other authors note (9, 10) that DA-like response properties can be separated from GABAergic-like responses merely on the basis of baseline firing rate (appealing for its agnosticism to cue responses); when selecting neurons firing fewer than 10 spikes/second (n=44), we again saw a phasic response profile with the same major effects (B/D discrimination t = 2.1, p> 0.05; A/C discrimination, t = 3.8, p<0.001; B-D & A-C correlation r =0.63, p<0.001).

Stimulus A and C also produced significant increases in the firing of putative dopamine neurons. The key comparison of the study was between these two responses, which was indeed, significantly greater for stimulus A. However, it is disconcerting how these responses spanned the response to stimulus D since the B to D and A to C comparisons appear (although it's not completely clear from the manuscript) to have been carried out independently without consideration of the variance of the responses to all of the stimuli. The concern is that if all the stimuli that were not directly paired with reward (A, C and D) were compared, significance would be lost. The authors should test whether the differences between responses to A and C are still significant if the analysis includes the responses to all of the stimuli (two-way ANOVA or comprehensive one-way ANOVA). At very least the authors should discuss if and why the responses to stimuli C and D are not more similar (e.g., more generalization to reward).

While we would have preferred lower neural responding for cue D, we feel strongly that the appropriate control comparison for cue A is cue C (in terms of overall number of exposures and previous presentation history in the context of reward). We agree that cue D might accumulate a bit of generalization to reward, as it has been presented in the context of reward for the previous 6 days (where A and C have never been presented in the context of reward), through mere exposure might alter the salience of cues, and studies have shown that subsets dopamine neurons track stimulus salience exclusively (4). To include responding to the conditioned cues in our analysis of the differences amongst cues A and C, we ran three additional ANOVAs: a two way “unrewarded cue” x time ANOVA (akin to analysis of voltammetry data testing the differences amongst the full dynamic cues responses) finding effects of cue (F = 29.0, df = 2, p<0.001) and time (F = 3.45, df = 200; p<0.001), a two way – “reward-predicting” x “preconditioning” ANOVA (where A/B, C/D are grouped for the first factor and A/C and B/D are grouped for the second) finding a significant effect of reward (F = 4.6, df = 1, p<0.05), and a comprehensive simple one way ANOVA across responses to all cues (F = 4.99, p<0.01) where A and C were significantly different (tukey-kramer, p<0.05).

References

1. J. L. Jones et al., Orbitofrontal cortex supports behavior and learning using inferred but not cached values. Science 338, 953 (2012).

2. H. M. Wied, J. L. Jones, N. K. Cooch, B. A. Berg, G. Schoenbaum, Disruption of model-based behavior and learning by cocaine self-administration in rats. Psychopharmacology 229, 493 (2013).

3. T. A. Stalnaker et al., Orbitofrontal neurons infer the value and identity of predicted outcomes. Nature Communications 5, 3926 (2014).

4. E. S. Bromberg-Martin, M. Matsumoto, O. Hikosaka, Dopamine in motivational control: rewarding, aversive and alerting. Neuron 68, 815 (2010).

5. Y. Takahashi et al., The orbitofrontal cortex and ventral tegmental area are necessary for learning from unexpected outcomes. Neuron 62, 269 (2009).

6. Y. K. Takahashi et al., Neural estimates of imagined outcomes in the orbitofrontal cortex drive behavior and learning. Neuron 80, 507 (2013).

7. M. Gallagher, R. W. McMahan, G. Schoenbaum, Orbitofrontal cortex and representation of incentive value in associative learning. Journal of Neuroscience 19, 6610 (1999).

8. M. A. McDannald, F. Lucantonio, K. A. Burke, Y. Niv, G. Schoenbaum, Ventral striatum and orbitofrontal cortex are both required for model-based, but not model-free, reinforcement learning. Journal of Neuroscience 31, 2700 (2011).

9. J. Y. Cohen, S. Haesler, L. Vong, B. B. Lowell, N. Uchida, Neuron-type-specific signals for reward and punishment in the ventral tegmental area. Nature 482, 85 (Feb 2, 2012).

10. N. K. Totah, Y. Kim, B. Moghaddam, Distinct prestimulus and poststimulus activation of VTA neurons correlates with stimulus detection. Journal of neurophysiology 110, 75 (Jul, 2013).